Application of intestinal microbiota in marine fish for assessing the toxicity of typical pollutants: a literature review

Feng Yunzhi 1 2
Xu Haolong 1 2 18868830390@126.com
Xu Guohua 1 2
http://orcid.org/0000-0002-6666-1480 Sun Dong 3
1 Ecological and Environmental Science and Research Institute of Zhejiang Province , Hangzhou , China
2 Zhejiang Key Laboratory of Ecological Environmental Damage Control and Value Transformation , Hangzhou , China
3 Key Laboratory of Marine Ecosystem Dynamics, Second Institute of Oceanography, Ministry of Natural Resources , Hangzhou , China
Fernandes Carlos Eurico
Electronic publication date: 2025 Oct 28
Publication date: 2025
Volume: 13
Electronic Location ID: e20248
Received 2025 Jun 4; Accepted 2025 Sep 25
Copyright: © 2025 Feng et al.
Copyright year: 2025
Copyright holder: Feng et al.
License: This is an open access article distributed under the terms of the Creative Commons Attribution License, which permits unrestricted use, distribution, reproduction and adaptation in any medium and for any purpose provided that it is properly attributed. For attribution, the original author(s), title, publication source (PeerJ) and either DOI or URL of the article must be cited.
License URL: https://creativecommons.org/licenses/by/4.0/

Keywords: Pollutants, Gut microbiota, Toxicity, Marine fish

Funding: National Key R&D Program of China 2023YFC2811305 Central Guiding Local Science and Technology Development Fund Program of China 2025ZY01090 Zhejiang Provincial Government Subsidy for Provincial Institutes and Research Institutes 0443202402029 This work is supported by the National Key R&D Program of China (2023YFC2811305), Central Guiding Local Science and Technology Development Fund Program of China (2025ZY01090) and the Zhejiang Provincial Government Subsidy for Provincial Institutes and Research Institutes (0443202402029). The funders had no role in study design, data collection and analysis, decision to publish, or preparation of the manuscript.

==============================
The widespread diffusion and dilution of pollutants in the ocean lead to prolonged exposure of marine organisms to low-concentration contaminated environments, raising growing concerns about the potential risks associated with chronic low-level pollution. The gut microbiota of fish plays a pivotal role in essential physiological processes, which are critical for host health. Therefore, the key microbes in the gut could serve as valuable biomarkers for assessing the toxic effects of pollutants. This article systematically reviews the structure and functions of marine fish gut microbiota, outlines the primary methodologies for assessing gut microbiota, and highlights the impacts of typical pollutants (including petroleum hydrocarbons, antibiotics, heavy metals, and microplastic) on the composition, functionality, and metabolic activities of marine fish gut microbiota. In the future, integrating multi-technology approaches to investigate the toxic mechanisms of pollutants on gut microbiota and their biodegradation pathways will represent a pivotal direction in marine ecotoxicology research.

Introduction

As the economy develops rapidly, environmental pollution has caused increasingly severe damage to marine ecosystems. The mortality effects of pollutants on marine organisms, such as the lethal concentration 50 (LC50) value, have garnered extensive attention and research. However, it is noteworthy that high-concentration pollutants typically occur only around pollution sources (Zhou & Zhang, 1989). With dispersing and diluting by ocean currents, pollutants in the environment exist at low concentrations in most cases, and would cause long-term threats to marine life across broad areas (Schwarzenbach et al., 2006; Tarigan et al., 2025). Such threats manifest in various physiological impacts, including impaired growth, development, metabolic processes, and reproductive functions of marine organisms, ultimately leading to population decline. Consequently, most traditional methods focusing solely on physicochemical detection of lethal concentrations are insufficient to fully assess the ecological risks of pollutants in marine environments. Increasing attention has been directed toward the toxicological effects of pollutants at low concentrations.

Fish occupy key roles in marine food web and have important economic value (Travers-Trolet et al., 2025). Various microbes live in the digestive tract of fish, where they play crucial roles in many physiological processes, including food digestion, nutrient absorption, and immune regulation, all of which are closely related to host health (Cerf-Bensussan & Gaboriau-Routhiau, 2010; Koch et al., 2018; Luan et al., 2023). Additionally, the gut microbiota metabolites can affect the liver and brain through the microbiota-gut-liver axis and the microbiota-gut-brain axis (Cryan et al., 2019; Morais, Schreiber & Mazmanian, 2021; Tripathi et al., 2018). The gut serves as the primary target site for most pollutants upon entering the body, where the toxic substances can disrupt the stable state of the microbial communities (Medina-Félix et al., 2023). Consequently, dynamics of gut microbiota can be used as a proxy to assess the fish health condition and the toxicity of pollutants. On the other hand, the intestinal tract harbors diverse probiotic communities capable of the biotransformation of pollutants through a variety of biocatalytic reactions, including hydrolysis, reduction, N-oxide cleavage, functional group removal, and denitrification. These microbes also enhance the host’s resistance to exogenous pollutants by regulating critical physiological processes (Claus, Guillou & Ellero-Simatos, 2016; Haiser & Turnbaugh, 2013). Research on probiotics contributes to the development of bioproducts aimed at regulating the gut microbiota of fish to improve their health (Hoseinifar et al., 2016; Wang et al., 2021, 2018). The unique conditions of marine environment results in differences in both the structure and biological activity of the metabolites synthesized by probiotics, compared to their terrestrial counterparts, thereby providing new directions for the development of bioproducts (Uniacke-Lowe et al., 2024). Considering the key role of gut microbiota to the host, the interaction between intestinal microbes and pollutants has emerged as a research focus. However, only a limited proportion of studies have addressed marine fish. Given the substantial differences between freshwater and marine environments, toxicological findings derived from freshwater fish studies should not be directly extrapolated to marine ecosystems (Jeon et al., 2010). In this review, we will make a systematic summary of the composition, function and research methods of marine fish gut microflora, as well as the effects of typical pollutants such as petroleum, antibiotics, heavy metals and microplastics on microflora. Such knowledge will provide reference and inspiration for further understanding of toxic effects and ecological risks of marine pollutants. The audience for this review includes researchers and scientists studying in marine ecotoxicology, aquatic gut microbiome studies, and marine environmental health science, particularly those focused on marine fish as a model organism.

Survey methodology

The Web of Science (WOS) Core Collection database and Google Scholar were primarily used for bibliographic searches in this study. First, to introduce and discuss the composition and function of marine fish gut microbiota, we used the following keywords: (“marine fish” AND (gut OR intestinal) AND (microbiota OR microflora OR “intestinal microbes” OR “intestinal microorganisms”). Then, to summarize the application of marine fish gut microbiota for assessing pollutant toxicity, we added the following keywords: “pollution” OR “pollutant” OR “contamination” OR “contaminant” OR “toxicity” OR “oil” OR “petroleum” OR “antibiotics” OR “microplastics OR “nanoplastics” OR “heavy metal”. Through manual proofreading, we incorporated studies based on the following inclusion criteria: the experimental animals used in these studies were marine fish, and studies discussing the effects of pollutants on the gut microbiota, including alteration in biodiversity, function, the associated bidirectional mechanisms and so on. The included studies were subsequently categorized based on types of pollutants. We also incorporated some articles on zebrafish, an important freshwater model fish, for improving the foundational basis of our work.

Composition and function of marine fish microbiota

The fish intestinal tract harbors a diverse microbial community comprising bacteria, archaea, fungi, viruses, and protists, with bacteria being the predominant group (Rombout et al., 2011). The intestinal microbiota of fish can be divided into two primary groups: indigenous microbes colonizing the mucosal surfaces and free-living allochthonous microbes. The resident ones predominantly consist of Proteobacteria, Firmicutes, Bacteroidetes, Fusobacteria, Actinobacteria, and Verrucomicrobia (Ringø et al., 2006). The diversity and community structure of fish intestinal microbiota are affected by many factors such as the host, sex, diet, and environment factors (Luan et al., 2023). Generally, the aerobic and facultative anaerobes from the phylum Proteobacteria are predominant in the fish gut (Rombout et al., 2011) compared to mammals like humans, primarily due to the typically higher oxygen within the fish intestinal environment (Singh et al., 2025). Significant variations of gut microbiota exist between marine fish and freshwater fish due to the distinct environment. Some marine herbivorous fish gut harbored more anaerobes from Firmicutes that aided fermentative digestion (Egerton et al., 2018). At the genus level, the gut microbiota is more distinct between marine fish and freshwater fish (Singh et al., 2025). For example, in the most common phylum Proteobacteria, freshwater fish often contain species such as Aeromonas, Pseudomonas, and Enterobacter (Deb, Das & Das, 2020; Givens et al., 2015), while marine fish usually harbor species such as Vibrio, Photobacterium, and Shewanella (Egerton et al., 2018; Givens et al., 2015).

The gut microbiota plays a crucial regulatory role in numerous key physiological processes of the fish, including nutrient absorption, development, immune defense, nervous system function, endocrinology and so on (Luan et al., 2023; Ou et al., 2021). Many bacteria genera, such as Bacillus, Vibrio, Photobacterium, Aeromonas, Flavobacterium, and Pseudomonas, may participate in the digestive process of fish gut (Ray, Ghosh & Ringø, 2012). In marine fish, several microbes found in the gut have specific functions. For instance, Aliivibrio are involved in bioluminescence (Klemetsen, Karlsen & Willassen, 2021) and Pseudoalteromonas have important function in interactions with the host (Drønen et al., 2022). However, many of these genera also contain pathogenic species (Egerton et al., 2018). The intestinal microbiome of fish is a complex and dynamic environment, where diverse microbial taxa maintain a state of dynamic equilibrium. Environmental stressors, exemplified by pollutant exposure, can disrupt the balance of the gut microbiota. Previous studies have suggested that some genera present in marine fish gut have the potential to degrade pollutants (Table 1). Theses bacterial genera were originally isolated from environmental samples and their pollutant-degrading capabilities are well-documented in these environments. However, direct evidence of their function in fish guts remains limited. In freshwater fish, some probiotic strains have been found to mitigate the intestinal toxicity of pollutants by reducing their accumulation in the gut or alleviating tissue oxidative stress and inflammation. For instance, Lactobacillus reuteri and Lactobacillus plantarum have demonstrated the capacity to alleviate the toxicity of lead (Giri et al., 2018) and cadmium (Zhai et al., 2017), respectively. Considering the strong variation between marine and freshwater environment, whether the presence and function of these species can be applied to marine fish was unknown. Research on these functional intestinal bacteria will provide fundamental support for developing biological agents and pharmaceuticals to enhance the resistance of fish to pollutants.

Table 1 Potential pollutant-degrading bacterial genera identified in environmental samples, with hypothesized relevance to marine fish gut microbiota.

Class	Genus	Pollutants	References related to their functions	References related to their presence in marine fish gut	
Alphaproteobacteria	Vibrio	Polycyclic aromatic hydrocarbons	Hedlund & Staley (2001)	Egerton et al. (2018)	
Alphaproteobacteria	Thalassospira	Polycyclic aromatic hydrocarbons	Kayama, Kanaly & Mori (2022)	Kang et al. (2021)	
Betaproteobacteria	Burkholderia	Aromatic hydrocarbons, antibiotics, heavy metals	Denef (2007), Jiang et al. (2008)	Wang et al. (2020)	
Betaproteobacteria	Acinetobacter	Aromatic hydrocarbons, heavy metals, alkane	Czarny et al. (2020), Révész et al. (2020)	Egerton et al. (2018)	
Gammaproteobacteria	Alteromonas	Selenium, tellurium	Reddy, Pathak & Nancharaiah (2023)	Egerton et al. (2018)	
Gammaproteobacteria	Alcanivorax	Alkane	Hara, Syutsubo & Harayama (2003)	Walter, Bagi & Pampanin (2019)	

Common methods used in investigating gut microbiota

Culture-dependent methods

Early investigations of fish intestinal microbiota were predominantly carried out by conventional microbial culture methods (Ringø & Birkbeck, 1999). These approaches enabled the identification and quantification of isolated strains, thereby revealing the composition and functional characteristics of intestinal microbes. Furthermore, using diverse selective media could obtain functional intestinal bacteria with unique physiological and biochemical properties. This methodological framework is essential for validating microbial functions, determining their mechanistic roles within the intestinal ecosystem (Ceja-Navarro et al., 2015), and subsequently developing novel microbial agents to enhance fish immunity (Dawood & Koshio, 2016). Whereas, 97% of marine intestinal bacteria are still uncultured due to their unique growth environment (Soh et al., 2024), indicating the limitations of traditional culture methods. Recently, the emerging culturomics provides a new tool for studying the function and role of intestinal microorganisms. Culturomics refers to an advanced methodology that employs various culture conditions to simulate the natural growth environment of microbes, thereby facilitating the cultivation of previously unculturable bacterial species. This approach integrates matrix-assisted laser desorption/ionization time-of-flight mass spectrometry (MALDI-TOF MS) and amplicon sequencing techniques for rapid bacterial identification and optimization of culture conditions. Through this innovative strategy, a substantial number of novel bacterial strains have been successfully isolated and characterized (Lagier et al., 2018). Currently, culturomics is mainly used in the study of human intestinal microorganisms. The primary limitations of this approach reside in its requirement for extensive testing and optimization of diverse culture conditions, coupled with high research cost (Lagier et al., 2018).

Molecular-based methods

(1) Amplicon sequencing

As the high-throughput sequencing and bioinformatics technology develop, the culture-independent molecular-based methods have greatly improved our understanding of fish gut microbiota. The amplicon sequencing technology has become the most common method to study the diversity and community structure of intestinal microbes (Ghanbari, Kneifel & Domig, 2015). This method enables rapid, efficient and cost-effective analysis of microbial diversity, community structure and their response to pollution in the fish gut, and enable identifying key bacteria with biomarker potential. The amplicon sequencing method is usually based on second-generation sequencing platforms such as 454 pyrosequencing and Illumina to sequence one or more variable regions of the marker gene (e.g., 16S rRNA for bacteria and archaea, as well as internal transcribed spacer (ITS) or 18S rRNA for fungi). However, due to the short sequencing read length of these second-generation platforms, the taxonomic identification results from sequence are usually limited to the genus or family level. The emerging third-generation sequencing platforms such as PacBio and Nanopore enable the full-length sequencing of the 16S/18S rDNA or ITS gene, and thus significantly improve the accuracy of species annotation. However, there are also problems such as higher cost and more sequencing error. Furthermore, the amplicon sequencing technology, which relies on polymerase chain reaction (PCR) amplification, is inevitably affected by methodological limitations including primer mismatch and interspecies variations in genomic copy numbers. These methodological limitations induce amplification biases, thereby compromising the accuracy of microbial diversity characterization and relative abundance quantification (Alberdi et al., 2018; Rathod & Silverman, 2025).

(2) Meta-omics

The rapid advancement of omics technologies has provided novel approaches for elucidating comprehensive functional profiles of gut microbial communities. Firstly, metagenomics offers significant advantages by directly performing large-scale random shotgun sequencing of the total genomic DNA extracted from samples. This methodological innovation enables accurate characterization of microbial community composition at the species level, and could effectively avoid amplification biases due to independent of primer selection and PCR amplification processes (Hugenholtz & Tyson, 2008). In addition, metagenomics can obtain the whole genomic information of the microbiota, so it can reveal the changes of functional genes in the community and identify key functional genes that are sensitive to pollutants. To further analyze the expression levels of functional genes, the metatranscriptome method is needed. Through mRNA isolation and enrichment, followed by reverse transcription into cDNA and subsequent high-throughput sequencing, transcriptomics enables the identification of active microbes within the community as well as the characterization of dynamic changes in expression levels of functional gene. This approach provides valuable insights into pollutant metabolic processes at the transcriptional level (Sorek & Cossart, 2010). When combined with metagenomics results, it is also possible to determine which bacteria in the community are transcribing and estimate the transcription rates. The primary limitations of this technology come from the inherent characteristics of mRNA, including its low abundance and susceptibility to degradation, which makes sample preparation process quite challenging. Furthermore, while high-throughput sequencing provides comprehensive profiling capabilities, it lacks precise quantitative accuracy. To obtain reliable quantification of taxonomic abundance and functional gene expression levels, complementary techniques such as quantitative real-time PCR (qPCR) and fluorescence in situ hybridization (FISH) are typically employed.

The information obtained from DNA/RNA is still partial, because gene transcripts can form a variety of proteins through different ways of splicing and post-translational modification (Cho, 2007). Since proteins are directly involved in multiple biological processes, understanding their types and functions are crucial for revealing the response of the organism to environmental changes. The metaproteomics approach works by separating and extracting proteins, mass spectrometry (MS) analysis, and database matching. This method helps reveal the protein expression patterns in microbial communities, providing deeper insights into both the toxic effects and the degradation mechanism of pollutants (Franzosa et al., 2015). Despite its potential, metaproteomics has limited application in fish gut microbiota research due to several challenges, including high costs, technical complexity, and the issue of host contamination (Ou et al., 2021).

Emerging after genomics, transcriptomics, and proteomics, metabolomics represents the most phenotype-proximal omics. This approach primarily employs nuclear magnetic resonance (NMR) and MS technologies to analyze the complete set of small-molecule metabolites (molecular weight <1,000 Da) within biological systems, including amino acids, sugars, vitamins, lipids, and so on (Nicholson, Lindon & Holmes, 1999). Metabolites are the end products of gene expression, subtle changes in genes and proteins may be amplified at the metabolic level. Therefore, metabolites detection techniques have the highest sensitivity for assessing functional changes. Moreover, the number of metabolites is less than that of proteins, contributing to lower technical difficulty than metaproteomics analysis. In current fish gut microbiome toxicology studies, metabolomics is typically used in conjunction with amplicon technology to analyze the interaction between gut microbiota and metabolic levels of host under pollutants (Duan et al., 2024; Wang et al., 2023; Zhang et al., 2022).

Gnotobiotic models

Gnotobiotic models refer to animals that are raised in germ-free environments or colonized with specific microbial species (Pham et al., 2008). These models serve as precise tools for investigating the bidirectional relationship between gut microbiota and host by eliminating interference from indigenous microbes. Gnotobiotic models have been established in several marine fish species, such as Atlantic cod (Gadus morhua L.) (Forberg, Arukwe & Vadstein, 2011), sea bass (Dicentrarchus labrax) (Dierckens et al., 2009), and Atlantic salmon (Salmo salar L.) (Gómez de la Torre Canny et al., 2023). These models are not only utilized to investigated the interactions between host and opportunistic pathogens (Li et al., 2015), but also enable evaluation of probiotic candidates (Aerts et al., 2018; Schaeck et al., 2017) and bioproducts (Yaacob et al., 2017). Future research with fish gnotobiotic models should integrate molecular-based methods to systematically map the bidirectional pollutant-microbiota interactions. By transplanting specific gut microbes into gnotobiotic fish models, the functions and mechanisms of key bacterial strains underlying pollutant toxicity and degradation could be validated, providing a theoretical foundation for pollution management in aquaculture.

Summary of methods

Culture-dependent methods and molecular-based approach are both important for current research on the toxicological effects of pollutants on fish gut microbiota. In practical investigations, the selection of appropriate methodologies should be guided by specific research objectives (Fig. 1, Table 2). For instance, the amplicon sequencing with mature technology and low cost is suitable for the analysis of large-scale samples, which is helpful to establish the risk assessment system of pollutants. The multi-omics methods, which is informative but costly, is suitable for in-depth exploration of the interaction between pollution and fish gut microbiota. The challenges for future multi-omics research include enhancing the quality of sequencing data, improving the accuracy and coverage of reference databases, and integrating large amounts of data from various omics. Additionally, in vitro culture experiments, and gnotobiotic fish models can be used for subsequent verification. The combination of these methods can provide a theoretical basis for the derivation of environmental and health criteria for pollutants as well as the development of bioremediation.

Figure 1 Summary of molecular-based methods for gut microbiota research.

Table 2 Comparison of molecular-based methods for gut microbiota research.

Methods	Objects	Advantages	Limitations	
Amplicon	Regions or full-length of the marker gene	It can obtain species composition and relative abundance. (Who are they?)	PCR-bias.	
Taxonomic identification are usually limited to genus or family level.	
Metagenomics	All genes	Taxonomic identification can be assigned to species and even strains level.	Higher sequencing throughput is required.	
		It can obtain composition and relative abundance of functional genes. (What can they do?)	It cannot distinguish between active and dormant/dead microbes.	
Metatranscriptomics	mRNA	It can obtain information of active microbes.
It can identify actively expressed genes and metabolic pathways. (What are they doing?)	Due to low abundance and poor stability of mRNA, preparation and preservation of samples are quite challenging.	
Metaproteomics	All protein	It can obtain the types and quantities of all proteins in the microbiota, and subsequent reveal protein expression patterns. (How do they function?)	The extraction and identification techniques are complex.	
It is difficult to eliminate host contamination.	
Metabolomics	All small-molecule metabolites	It can obtain the types and quantities of all small-molecule metabolites (molecular weight < 1,000 Da), reflecting the metabolic state. (What is the result?)	Only small-molecule metabolites can be measured.	
It is difficult to determine the specific biological origin and action pathway of metabolites	

Effects of typical pollutants on gut microbiota of marine fish

Petroleum pollutants

Oil spill accidents resulting from marine transportation and offshore oil extraction, coupled with substantial discharges of oil-containing wastewater, have posed significant threats of petroleum pollution to marine environments (Chen et al., 2018; Yu, Xia & Du, 2024). Upon entering the marine ecosystem, petroleum initially forms surface oil slicks that impede air-water exchange, subsequently undergoing dispersion, migration, and weathering processes driven by wind and ocean currents. This leads to widespread exposure of marine organisms to low-concentration petroleum pollutants in the water. Petroleum represents a highly complex mixture primarily composed of toxic components, such as saturated hydrocarbons, aromatic hydrocarbons, and heavy metals (Jovančićević et al., 2007). Exposure of fish intestinal systems to either crude petroleum or its secondary metabolites transformed by the liver can induce significant shifts in gut microbiota composition. This community succession is predominantly characterized by the enrichment of microbes capable of utilizing petroleum components as carbon sources (Améndola-Pimenta et al., 2020; Cerqueda-García et al., 2020; Walter, Bagi & Pampanin, 2019), which causes harm to the functional structure of gut microbiota. There are several studies focus on the response of marine fish gut microbiota to petroleum pollution as follows. Bagi et al. (2018) conduct a 28-day chronic exposure experiment to Atlantic cod by simulating the leakage scenario of low-concentration crude oil (0.01, 0.05, 0.1 mg/L). The result showed that the morphological parameters of fish in all experimental groups were not significantly different from the control group. However, the intestinal microflora of Atlantic cod exposed to medium (0.05 mg/L) and high (0.1 mg/L) concentrations of dispersed crude oil showed significant changes, reflecting the sensitivity of intestinal microflora to the toxicity of low-concentration pollutants. Specifically, the relative abundance of Deferribacteres in the gut increases with oil exposure levels, indicating it can be used as a potential biomarker of oil exposure in Atlantic cod. Spilsbury et al. (2022) fed juvenile Asian seabass (Lates calcarifer) with diets spiked with petroleum hydrocarbons for 33 days. They presume that the enrichment of Photobacterium in the intestine microbiota was proportional to the dietary exposure concentration of polycyclic aromatic hydrocarbons (PAHs). Consequently, Photobacterium was identified as a potential biomarker genus of PAHs exposure. Beyond amplicon sequencing, a comprehensive study using multi-omics techniques was conducted on juvenile Atlantic cod exposed to 0.05 mg/L dispersed crude oil (Magnuson et al., 2023). After 28 days of exposure, a metagenomic tool predicted significant changes in gut microbial functions, particularly affecting energy and carbohydrate metabolism, fatty acid and amino acid biosynthesis, as well as cellular structure formation. Metatranscriptomics data showed differential expression of 58 genes after 7 days of exposure. Notably, the transcriptomic response in intestinal contents was more sensitive, showing significant changes after just 1 day of exposure, reflecting the dynamic interactions between gut microbiota and host. Another study examined the effects of water accommodated fraction of a light crude oil (5% and 10% w/w) on the lined sole (Achirus lineatus) from the Gulf of Mexico over 28 days (Cerqueda-García et al., 2020). The research revealed an enrichment of menaquinone and demethylquinone biosynthesis pathways in the gut, suggesting these changes might shift the intestinal redox state to be more oxygen-limited. In summary, analyzing fish gut microbiota opens new possibilities for developing biomarkers of petroleum pollutant exposure. Multi-omics technologies can uncover the functional and metabolic impacts of petroleum pollutants, contributing to the construction of highly sensitive and specific toxicity assessment framework.

Antibiotics

Antibiotics, defined as natural or synthetic compounds capable of killing bacteria or inhibiting bacterial growth, are widely used in disease treatment. However, their extensive misuse, combined with low bioavailability, high water solubility, and environmental persistence, pose significant ecological risks. When introduced into organisms, antibiotics are primarily absorbed through the intestinal tract, where they can damage intestinal tissues and eliminate substantial populations of sensitive gut microorganisms (Morgun et al., 2015; Zhou et al., 2018). Additionally, this exposure may lead to the development of antibiotic resistance within the gut microbiota (Francino, 2016). Wang et al. (2023) found that short-term (9 days) exposure to the fluoroquinolone antibiotic enrofloxacin (5–500 µg/L) increased the ratio of firmicutes abundance to Bacteroidetes abundance (F:B ratio) in the gut microbiota of marine medaka (Oryzias melastigma). The elevated (F:B ratio) is seen as an indicator of gut microbiota disorders (Shen et al., 2018). At the genus level, the relative abundance of several pathogenic bacteria in the genera Flavobacteria and Vibrio, as well as the probiotic bacterium Shewanella decreased. It reflects the imbalance of intestinal microbiota under enrofloxacin exposure. Moreover, enrofloxacin exposure led to significant enrichment of antibiotic resistance genes, as well as further negative effects on liver metabolism via the gut-liver axis. He et al. (2022) reported the intergenerational effect of the sulfanilamide antibiotic sulfadiazine on the intestinal flora of marine medaka. After the parents were exposed to sulfadiazine (5 mg/g) through diet for 30 days, the diversity of intestinal microbiota in the offspring was significantly reduced, and the relative abundance of Cyanobacteria was significantly increased. These intergenerational effects reflect the long-term risk of antibiotics to marine organisms.

Antibiotics are also extensively used in aquaculture for disease treatment, potentially impacting fish gut health. Sumithra et al. (2022) administered florfenicol, a chloramphenicol-class antibiotic, at a therapeutic dose of 10 mg/kg in feed to snubnose pompano (Trachinotus blochii) for 10 days, following by withdraw. By means of conventional culturing method, they found a 2-log reduction in the total viable bacteria compared to the control fish gut, which restored at the 10th-day post-withdrawal. The 16S amplicon sequencing approach revealed significantly reduced gut microbiota diversity, with increased relative abundance of Proteobacteria and decreased relative abundance of Firmicutes, following by their restoration at 10th-day post-withdrawal. However, the relative abundances of some specific taxa, including Vibrionaceae and Enterobacteriaceae, remained elevated compared to controls after 15th-day post-withdrawal. Additionally, the results indicated that therapeutic exposure to florfenicol neither promoted irreversible enrichment of resistant bacteria nor caused irreversible increase in antibiotic resistance in the fish gut. Sumithra et al. (2024) also evaluated the effects of oxytetracycline, another antibiotic for therapeutic use, on snubnose pompano over a 10-day exposure period. They found that the perturbation of gut microbiota recovered in 5–15 days of withdraw, without inducing antibiotic resistance. These results indicated that the toxic effects of antibiotics at a therapeutic dose on fish gut microbiota are partially reversible. In summary, dynamic monitoring of fish gut microbiota facilitates a comprehensive assessment of antibiotic exposure risks. Further studies are needed to investigate the dose-time-effect relationships of antibiotics on the gut microbiota of marine fish, and subsequently establish environmental and health-based criteria. These researches will provide scientific guidance for more rational antibiotic use in mariculture.

Heavy metals

Heavy metal pollution has become one of the most severe global environmental challenges. Heavy metals have extremely wide sources, which not only involves natural activities such as submarine volcanic eruptions and crustal movements, but also includes human-induced leaks from industries like oil, mining, agriculture, and chemicals engineering (Abd Elnabi et al., 2023). Heavy metals have strong bioaccumulation properties. Once they enter the intestinal tract, they can damage the intestinal mucosa, alter the composition and metabolic characteristics of the intestinal flora, and thereby trigger various metabolic diseases (Duan et al., 2020). To the best of our knowledge, research on the interaction of heavy metals and the intestinal flora of marine fish is relatively limited. Liu et al. (2024) reported that after the red-spotted pufferfish (Takifugu rubripes) exposed to copper (50–500 μg/L) for 3 days, the relative abundances of various pathogenic bacteria in the fish gut increased, while the relative abundances of some probiotics decreased. The correlation analysis results of 16S amplicon and metabolomics suggest that the key response bacteria exposed to high copper concentration may up-regulate the level of gut metabolites in the tryptophan metabolic pathway, leading to intestinal metabolic dysbiosis. Duan et al. (2024) conducted a 14-day exposure experiment using 1 μg/L lead on the grouper (Epinephelus fuscoguttatus), an economically important marine fish species. The results demonstrated that lead exposure induced gut microbiota dysbiosis characterized by increased relative abundances of Proteobacteria and Actinobacteria, as well as an elevated F:B ratio. Furthermore, significant correlations were observed between the relative abundances of specific bacterial genera and hepatic physiological metabolites, suggesting that lead stress negatively affects the physiological homeostasis of E. fuscoguttatus through the gut-liver axis. Wang et al. (2020) investigated the response of combined exposure to cadmium (10 μg/L), lead (50 μg/L), and zinc (100 μg/L) on the gut microbiota of marine medaka for 1 month, revealing significant gender-specific differences. Compared to females, males exhibited more pronounced alterations in metabolic pathways, including carbohydrate metabolism, lipid metabolism, and biodegradation of xenobiotics. These results indicate greater sensitivity of male gut microbiota to heavy metals, potentially mediated by sex hormones (Mueller et al., 2006). However, the underlying mechanisms of gender-specific responses to heavy metal toxicity require further elucidation.

Microplastics

Microplastics, defined as plastic particles with diameters <5 mm (Thompson et al., 2004), have been detected in marine environments worldwide, exerting a substantial threat on marine biota (Ugwu, Herrera & Gómez, 2021). These particles undergo gradual fragmentation into smaller sizes through photodegradation and weathering processes in the ocean (Ryan et al., 2009), yet resist complete degradation. Hence, microplastic pollution has emerged as one of the most pressing environmental concerns. Due to their small size, microplastics are readily ingested by fish and accumulate in their gastrointestinal tracts (Lusher, McHugh & Thompson, 2013; Norhazwani et al., 2021). Such accumulation can disrupt gut microbial communities, with the impact degree influenced by particle size and exposure periods. Kang et al. (2021) studied the toxicity of two particle sizes of polystyrene microplastics (10 µg/mL) to juvenile marine medaka after 24-h acute exposure. The 16S amplicon results revealed a significant decrease in the relative abundance of Bacteroidetes and the probiotic genus Shewanella in the gut, while the relative abundance of Thalassospira, known for its ability to degrade various PAHs (Table 1), showed a marked increase. This suggests that Thalassospira, may be involved in the degradation of plastics. In summary, the acute exposure to microplastics caused disruption of intestinal flora, and microplastics with a diameter of 45 μm had a greater effect than nanoplastics with a diameter of 50 nm. Gu et al. (2020) conducted a 14-day chronic exposure experiment using 100 nm polystyrene nanoplastics (5.50 × 10−6 – 5.50 × 10−1 µg/mL) on juvenile large yellow croaker (Larimichthys crocea). The amplicon sequence results showed that the relative abundances of Bacteroidetes and Firmicutes in the gut generally increased. While Proteobacteria showed a downward trend, indicating that the growth of juvenile fish may be inhibited by nanoplastics. At the genus level, the proportion of the potential pathogenic bacteria Alistipes and Parabacteroide increased, posing a threat of microplastics to juvenile fish health. Conversely, the rise in the probiotic Lactobacillus may represent a compensatory response to counteract this threat, demonstrating the self-regulatory capacity of gut microbiota in response to nanoplastic exposure. Yao et al. (2024) exposed golden pompano (Trachinotus blochii) to 5 μm polystyrene microplastics (10–1,000 µg/L) for 14 days. The results of amplicon sequencing and metabolomics revealed both dysbiosis and metabolic alterations (involving lipids, glucose, and amino acids). These disruptions impaired the digestion and absorption functions and may lead to many diseases of the host. Feng et al. (2021) found that after feeding 2 and 200 μm polystyrene (3 µg/mg) to marine medaka for 28 days, the carbon/nitrogen/phosphorus/sulfur metabolic pathways of gut microorganisms were changed, which is revealed by metagenomic analysis. Moreover, the strong correlation between gut microbiota dynamics and hepatic transcriptomic response demonstrated that polystyrene exposure may affect host health by disturb the gut-liver axis. Wen et al. (2024) found that after marine medaka being exposured to low concentrations of polyethylene and polylactic acid (0.2 µg/mL) for 60 days, the antibacterial Streptomyces was significantly enriched in the gut, but the overall community structure of intestinal flora showed no significant difference to the control group. These findings indicated that although long-term exposure to low-concentration microplastics does not induce overall gut microbiota dysbiosis, the stress response of several bacterial taxa can serve as early warning signals for chronic toxic effects. Given the persistent nature of microplastics, their long-term accumulation in the environment may exert prolonged impacts on marine organisms throughout their life cycles and even across generations. Therefore, future research should prioritize investigating the chronic toxicity of microplastics and their transgenerational effects on the gut microbiota of marine fish.

Summary

The toxic effects of typical pollutants on marine fish gut microbiota are summarized in Table 3. Generally, these effects are manifested in two aspects as below: first, direct effects, where pollutants accumulate in the intestines, leading to changes in the composition, function, and metabolic pathways of the gut microbiota; second, indirect effects, where pollutants damage the intestinal mucosa, making microbial metabolites affect host health through the gut-liver axis or gut-brain axis. Under pollutant-induced stress, the gut microbiota can self-regulate to some extent. Various gut microbes have been found to possess the ability to mitigate the toxicity of pollutants, reducing their adverse effects on host health.

Table 3 Effects of typical pollutants on marine fish gut microbiota.

Types	Pollutants	Model organisms	Administered way	Main effects on gut microbiota
composition	Main effects on function and metabolic profiles	References	
Petroleum pollutant	Dispersed crude oil	Atlantic cod (Gadus morhua)	0.01, 0.05, and 0.1 mg/L; 28 d	0.05 and 0.1 mg/L groups: Bacteroidetes↓, Deferribacteres↑, Porphyromonadaceae↓, Rikenella↓, Ruminococcaceae↓, Alistipes↓, Clostridiales↓
0.01 mg/L group: no significant effect	–	Bagi et al. (2018)	
	Crude oil and fuel oil	Juvenile Asian seabass (Lates calcarifer)	1% w/w in diet; 33 d	Photobacterium↑, Vibrio↑	–	Spilsbury et al. (2022)	
	Dispersed crude oil	Juvenile Atlantic cod	0.05 mg/L; 28 d	Aliivibrio↑, Mycoplasma↑, Photobacterium↑	58 differentially expressed genes related to DNA, RNA, and ATP binding, translation, signal transduction, and Wnt signaling pathway	Magnuson et al. (2023)	
	Water accommodated fraction (WAF) of crude oil	Lined sole (Achirus lineatus)	5% and 10% w/w; 28 d	Bacteria associated to hydrocarbon degradation↑, such as Acinetobacter johnsonii, Alcanivorax diselolei, and Sneathiella chungangensis	Menaquinone and demethylquinone biosynthesis pathways↑	Cerqueda-García et al. (2020)	
Antibiotics	Enrofloxacin	Marine medaka (Oryzias melastigma)	5 and 500 µg/L; 9 d	Escherichia↑, Epulopiscium↑, Acinetobacter↑, Apibacter↑, Vibrio↓, Halioglobus↓, Shewanella↓, Flavobacterium↓, Haliea↓	Antibiotic resistance genes↑; Hepatic metabolism disorder associated with intestinal flora dysbiosis	Wang et al. (2023)	
	Sulfadiazine	Marine medaka	5 mg/g in diet; 30 d (Parental exposure)	Verrucomicrobia↓, Cyanobacteria↑	For female: Carbohydrate metabolism↓ For male: nine pathways significantly changed (e.g., Carbohydrate metabolism↓, lipid metabolism↓, glycan synthesis, metabolism↑)	He et al. (2022)	
	Florfenicol	Juvenile snubnose pompano (Trachinotus blochii)	10 mg/kg in diet; 10 d, followed by withdraw for 15 d	Restore after withdraw: Proteobacteria↑, Euryacrcheota↓, Firmicutes↓ Enterovibrio↑, Vibrio↑, Pseudomonas↑, Shewanella↑ Remain after withdraw: Serratia↓, Acinetobactera↓	Kanamycin resistant microbes↑, multidrug resistance encoding genes↑ (restored after withdraw)	Sumithra et al. (2022)	
	Oxytetracycline	Juvenile snubnose pompano	80 mg/kg in diet; 10 d, followed by withdraw for 15 d	Firmicutes↓, Actinobacteria↓, γ-Proteobacteria↓, Vibrio↓, Solirubrobacteriales↑, Mycoplasma↑, Mycoplasma↓kanamycin and ampicillin-resistant cultivable bacteria↓ (restored after withdraw)	Energy metabolism pathway↓	Sumithra et al. (2024)	
Heavy metal	Copper (Cu)	Red-spotted pufferfish (Takifugu rubripes)	50, 100, and 500 µg/L; 3 d	500 µg/L group: Granulicella↑, Family_XIII_AD3011_group↓ 100 µg/L group: Rothia↑, Corynebacterium↑, Lactobacillus↓, Clostridium_sensu_stricto_1↓ 50 µg/L group: Lachnoclostridium↑, Bergeyella ↑, Actinobacillus↑, Butyrivibrio↓, Ruminococcus_torques_group↓	Significant correlation between the changes of key gut bacteria and gut metabolites related to energy and immunity (e.g., the arginine biosynthesis pathway and tryptophan metabolism pathway)	Liu et al. (2024)	
	Lead (Pb)	Grouper (Epinephelus fuscoguttatus)	1 µg/L; 14 d	Proteobacteria↑, Actinobacteria↑, Streptococcus↑, Bacteroidales S24–7 group↑, Ruminococcaceae UCG-005↑, Ruminococcaceae UCG-014↑, Oscillibacter↑, Tenacibaculum↓, Bdellovibrio↓	Functions of primary bile acid biosynthesis and glycosphingolipid biosynthesis↑; Significant correlation between the changes of key gut bacteria and the physiological indexes and metabolites in the liver	Duan et al. (2024)	
	Cadmium (Cd), Lead (Pb), Zinc (Zn)	Marine medaka	Combined exposure of Cd 10 µg/L, Pb 50 µg/L, and Zn 100 µg/L; 1 month	For female: Bacteroidetes↑, Verrucomicrobia↑, Lachnoclostridium-10↑, Ruminococcaceae↑, Lactobacillus↑, Burkholderiales↑, Pseudomona↑ For male: Cyanobacteria↑, Aurantimonadaceae↑, Rhizobium↑, Rhizobiaceae↑, Paracoccus↑, Methylobacterium↑, Methylobacteriaceae↑, Aurantimonadacea↑	Males exhibited more pronounced alterations in metabolic pathways than females, including carbohydrate metabolism, lipid metabolism, and biodegradation of xenobiotics	Wang et al. (2020)	
Microplastics	Polystyrene (45 µm, 50 nm)	Juvenile marine medaka	10 µg/mL; 24 h	Bacteroidetes↓, Vicingus↓, Shewanella↓, Lewinella↑, Pseudomonas↑, Thalassospira↑, Parahaliea↑	–	Kang et al. (2021)	
	Polystyrene (100 nm)	Juvenile large yellow croaker (Larimichthys crocea)	5.50 × 10−6, 5.50 × 10−3, and 5.50 × 10−1 µg/mL; 14 d	Bacteroidetes↑, Firmicutes↑, Proteobacteria↓, Alistipes↑, Parabacteroide↑, Lactobacillus↑	–	Gu et al. (2020)	
	Polystyrene (5 µm)	Golden pompano (Trachinotus blochii)	10, 100, and 1,000 µg/L; 14 d	Mycoplasma↓	Disruptions in lipid metabolism, carbohydrate metabolism, and amino acid metabolism; activated hormones, cell growth, signaling pathways, and disease-related pathways.	Yao et al. (2024)	
	Polystyrene (2 µm, 200 µm)	Marine medaka	3 µg/mg in diet; 28 d	Bacteroidetes↑, Planctomycetes↑, Motilimonas↓, Propionigenium↓, Aliivibrio↓, Oleibacter↓, Pseudophaeobacter↑, Bythopirellula↑, OM60 [NOR5] clade↑, Rhodopirellula↑, Winogradskyella↑, Rubripirellula↑, Crocinitomix↑, Actibacter↑, Cryomorpha	Carbon degradation/fixation activities↑, stress-related genes↑, changes of the nitrogen/phosphorus/sulfur metabolic pathways; Significant correlation between the changes of key gut bacteria and the metabolitic pathways of the liver	Feng et al. (2021)	
	Polyethylene and polylactic acid	Marine medaka	0.2 µg/mL; 60 d	Streptomyces↑	–	Wen et al. (2024)	

Conclusions and future prospects

The microbes in marine fish exhibit various diversity and functions and plays a crucial role in maintaining host homeostasis. Multiple pollutants can exert effects on the gut microbiota, thereby impacting the health of marine fish. With advancements in multi-omics technologies, the bidirectional interactions between marine pollutants and the fish gut microbiota have been increasingly investigated. However, current toxicological studies on marine fish gut microbiota still have some limitations: (1) Research are mostly limited to descriptive analysis based on 16S amplicon sequencing results. Since fish gut microbes have significant individual variability (Clements et al., 2014), this will make it difficult to accurately attribute variation in gut microbiota to pollutants exposure (Chi et al., 2021); (2) The toxicity endpoints and safe concentrations of most marine pollutants remain largely unconfirmed up to now. Meta-omics techniques can identify toxicity endpoints more sensitive than traditional methods (Simmons et al., 2015). However, a critical gap persists in translating these sensitive indicators, such as specific microbial species or functional genes, into validated biomarkers. Current experiments based on meta-omics typically employ restricted concentration gradients. This limitation impedes the establishment of robust dose-response relationships, leaving the feasibility of these potential biomarkers uncertain.

Future research should integrate culture-based experiments, multi-omics technologies, and gnotobiotic fish models to thoroughly investigate the effects of pollutants on gut microbiota function and metabolism. Key focuses include elucidating the dose-time-effect relationships of pollutants, exploring sex-specific differences and transgenerational effects in toxicity, and developing gut probiotics capable of mitigating pollutant-induced harm (Fig. 2). These efforts will comprehensively reveal the potential impacts of pollutants on host health and marine ecosystems, while supporting sustainable practices in marine aquaculture.

Figure 2 Summary of toxicity effect of typical pollutants on marine fish gut microbiota and future research perspectives.

The authors acknowledge the AI tools (DeepSeek and ChatGPT 4.0) for language polishing and grammatical refinement in the preparation of our manuscript.

Additional Information and Declarations

Competing Interests

Dong Sun is an Academic Editor for PeerJ.

Author Contributions

Yunzhi Feng conceived and designed the experiments, analyzed the data, prepared figures and/or tables, and approved the final draft.

Haolong Xu conceived and designed the experiments, authored or reviewed drafts of the article, and approved the final draft.

Guohua Xu performed the experiments, authored or reviewed drafts of the article, and approved the final draft.

Dong Sun conceived and designed the experiments, authored or reviewed drafts of the article, and approved the final draft.

Data Availability

The following information was supplied regarding data availability:

This study is a literature review.

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
