# Peer review of "Application of intestinal microbiota in marine fish for assessing the toxicity of typical pollutants: a literature review"

_PeerJ, doi:10.7717/peerj.20248_

## Round 0.1 · original submission · Major Revisions

Dear Dr. Feng,

I'm sending you the reviewers' suggestions. Your manuscript has potential for publication; however, please consider adjusting it by responding to the attached comments.

Best regards

Reviewer 1 ·

Basic reporting

The manuscript “Application of intestinal microbiota in marine fish for assessing the toxicity of typical pollutants: a review” is a systematic review focusing on the utilization of marine fish gut microbiota to evaluate the toxicity of common pollutants like petroleum hydrocarbons, antibiotics, heavy metals, and microplastics. The review discusses the gut microbiota's composition and function, the methodologies employed in research, and the impact of these pollutants on both the microbiota and the health of fish.

Experimental design

The review is basic, focusing solely on summarizing the findings of existing studies without offering any personal insights. The authors did not delve deeply into the suggested topic of using intestinal microbiota in marine fish to evaluate the toxicity of common pollutants.

In my view, substantial enhancements are required for this manuscript to be deemed suitable for publication.

Validity of the findings

The authors did not reference the current literature in this review. Instead, they relied on older literature, much of which is over 10 years old. This trend is consistent throughout the manuscript. In a review article, it is expected to primarily include recent advancements (within the last 5 years) on the topic.

Additional comments

The manuscript “Application of intestinal microbiota in marine fish for assessing the toxicity of typical pollutants: a review” is a systematic review focusing on the utilization of marine fish gut microbiota to evaluate the toxicity of common pollutants like petroleum hydrocarbons, antibiotics, heavy metals, and microplastics. The review discusses the gut microbiota's composition and function, the methodologies employed in research, and the impact of these pollutants on both the microbiota and the health of fish.
The review is basic, focusing solely on summarizing the findings of existing studies without offering any personal insights. The authors did not delve deeply into the suggested topic of using intestinal microbiota in marine fish to evaluate the toxicity of common pollutants.
In my view, substantial enhancements are required for this manuscript to be deemed suitable for publication.
Introduction
- In line 34, the authors stated: “The lethal effects…” What would the effects be? It would be interesting to exemplify them here.
- Line 95: Regarding the section Composition and function of marine fish microbiota: This topic is currently being extensively researched by scientists worldwide. However, the authors did not reference the current literature in this section. Instead, they relied on older literature, much of which is over 10 years old. This trend is consistent throughout the manuscript. In a review article, it is expected to primarily include recent advancements (within the last 5 years) on the topic.
- In line 110, the authors start the sentence by stating: The microbes in fish gut provide various function…but they proceed to discuss studies that are focused on humans or mammals. It is crucial to include research specifically related to fish in order to accurately report these functions.
- In line 225, the authors stated: Gnotobiotic models have been established in several marine fish species, including Nile tilapia. Why do the authors refer to Nile tilapia (Oreochromis niloticus) as a marine species?
-
- Line 297 and 365: Antibiotics and Microplastics
- One of the most crucial sections to investigate, yet the authors either neglected or failed to emphasize the topic in the manuscript…at least with regard to the title of the manuscript.

Reviewer 2 ·

Basic reporting

The review aims to provide an overview of research on marine fish gut microbiota, with a focus on assessing the impacts of environmental pollution on fish. This is an important topic, especially as environmental safety assessments are increasingly adopting new approach methodologies. Highlighting recent, relevant literature adds value to the research field and aligns with the journal's scope. A recent, thorough review on a similar topic, “See, M. S., X. L. Ching, S. C. Khoo, S. Z. Abidin, C. Sonne and N. L. Ma (2025). "Aquatic microbiomes under stress: The role of gut microbiota in detoxification and adaptation to environmental exposures." Journal of Hazardous Materials Advances 17: 100612,” has been published. However, this review has a slightly different focus and does not outline the specific needs and goals of environmental toxicity and safety assessments, which is the main aim of the current manuscript. Therefore, this manuscript features novel and unique aspects that could be emphasized further to enhance its insightfulness and benefit to readers.

Experimental design

The content of Table 3 and the section “Effects of typical pollutants on gut microbiota of marine fish” pertain to the review topic and include valuable information about relevant studies on marine fish. However, the section on molecular-based methods for gut microbiota research actually discusses gut microbiota research in general, not specifically in marine fish. Therefore, either the title of this section should be changed to better match its content, or the section should be rewritten to focus on marine fish gut microbiome research. For example, the caption of Figure 1, “Summary of molecular-based methods for gut microbiota research,” more accurately describes the figure and the related section of the paper. Likewise, the caption of Table 2, “Comparison of molecular-based methods for gut microbiota research,” correctly reflects the table’s content, which is about gut microbiota research in general, not specific to marine fish. Also, in the Advantages column of Table 2, it is unclear what “How to do?” and “What are achieved?” mean. Please rephrase these for clarity and correct grammar.

Table 1. “Several pollutant-degrading genera in marine fish gut microbiota”. This caption is misleading because there is no direct evidence that these pollutant-degrading bacteria actually live in the guts of marine fish. The studies listed in the table have isolated these bacterial strains or communities from sediments or soils, rather than from the guts of marine fish. This point should be clearly stated in the paper. The authors make hypothetical assumptions that the gut microbiota of fish might have similar functions; however, there is no experimental evidence to support this idea.

There is an unusual subsection “Audience” included in the manuscript. I recommend removing it.

The first paragraph of the introduction lacks references. Some statements require support with citations.

In the section “Composition and function of marine fish microbiota,” it is unclear which information is specific to marine fish. Some statements are based on data from the human gut microbiota. It has not been specified which statements concern marine fish, freshwater fish, and mammals. As a result, the section's focus is unclear. Please rewrite this section by highlighting the differences between the gut microbiota of marine fish and that of other organisms. Discuss whether these differences are significant enough to prevent extrapolating research results from other species to marine fish (e.g., would data from freshwater fish not apply to marine fish). Here are some references that might help the authors in this task:
Ou W, Yu G, Zhang Y, and Mai K. 2021. Recent progress in the understanding of the gut microbiota of marine fishes. Marine Life Science & Technology 3:434-448. 617 10.1007/s42995-021-00094-y

Binoy Kumar Singh, Kushal Thakur, Hishani Kumari, Danish Mahajan, Dixit Sharma, Amit Kumar Sharma, Sunil Kumar, Birbal Singh, Pranay Punj Pankaj, Rakesh Kumar, A review on comparative analysis of marine and freshwater fish gut microbiomes: insights into environmental impact on gut microbiota, FEMS Microbiology Ecology 101(1), 2025, 169, https://doi.org/10.1093/femsec/fiae169

Some references cited in the text are missing from the reference list at the end, such as those in lines 125-126 and 133. Please review all references and update the reference list.

Additionally, while searching for similar literature and reviewing the cited papers, I found relevant studies that the authors have not included in their review. Sometimes, changing keywords specifically related to the topic can help find additional relevant papers. For example:
Najafpour, B., P. I. S. Pinto, E. C. Sanz, J. F. Martinez-Blanch, A. V. M. Canario, K. A. Moutou and D. M. Power (2023). "Core microbiome profiles and their modification by environmental, biological, and rearing factors in aquaculture hatcheries." Marine Pollution Bulletin 193: 115218.

Rimoldi, S., K. F. Quiroz, V. Kalemi, S. McMillan, I. Stubhaug, L. Martinez-Rubio, M. B. Betancor and G. Terova (2025). "Interactions between nutritional programming, genotype, and gut microbiota in Atlantic salmon: Long-term effects on gut microbiota, fish growth and feed efficiency." Aquaculture 596: 741813.



Validity of the findings

The authors should revise the sections “Composition and function of marine fish microbiota” and “Common methods used in investigating fish gut microbiota” to clearly indicate which statements pertain to marine fish and which to other organisms. Currently, there is a mismatch between the titles and the content of these subsections. The content should focus more on marine fish and include comparisons of their gut microbiota with that of other organisms. The review should provide insight into what makes the marine fish microbiota unique, highlighting its importance in environmental toxicology.

The conclusions currently appear to introduce new information, which they should not. The first paragraph of the Conclusions fits better in the subsection “Effects of typical pollutants on gut microbiota of marine fish." Conclusions should summarize and reflect on the previous subsections of the review.

Additional comments

If the authors are willing to substantially revise the review, it will be a valuable contribution to the field of environmental toxicology.

---

## Round 0.2 · accepted · Accept

Dear Dr. Feng,

Thank you for submitting your review to PeerJ. I congratulate you on your effort and on the acceptance of your manuscript for publication.

Reviewer 1 ·

Basic reporting

The revised manuscript has been significantly improved.
I believe the manuscript meets the minimum quality required for publication.
I rest my case.

Experimental design

ok

Validity of the findings

ok

Additional comments

The revised manuscript has been significantly improved.
I believe the manuscript meets the minimum quality required for publication.
I rest my case.

Reviewer 2 ·

Basic reporting

No comments

Experimental design

No comments

Validity of the findings

No comments

Additional comments

The authors have adequately addressed the reviewers' comments and made the necessary changes in the manuscript. The manuscript is now ready for publication.